# Alterations of Matrisome Gene Expression in Naturally Aged and Photoaged Human Skin In Vivo

**DOI:** 10.3390/biom14080900

**Published:** 2024-07-25

**Authors:** Yan Yan, Hehui Quan, Chunfang Guo, Zhaoping Qin, Taihao Quan

**Affiliations:** 1Department of Dermatology, University of Michigan Medical School, Ann Arbor, MI 48109, USA; yanyan@psh.pumc.edu.cn (Y.Y.); cfguo@umich.edu (C.G.); qinzp@med.umich.edu (Z.Q.); 2Lenox Hill Hospital, 100 E 77th St., New York, NY 10075, USA; hehuiquan@gmail.com

**Keywords:** matrisome, ECM, collagen, skin aging

## Abstract

The main component of human skin is a collagen-rich extracellular matrix (ECM), known as the matrisome. The matrisome is essential for maintaining the structural integrity and mechanical properties of the skin. Recently, we reported notable decreases in matrisome proteins in natural aging and photoaging human skin. This study aims to investigate the mRNA expression of the core matrisome proteins in human skin, comparing young versus aged and sun-protected versus sun-exposed skin by quantitative real-time PCR and immunostaining. Our findings reveal a notable decrease in core matrisome transcription in aged skin. The mRNA expression of the core matrisome, such as collagen 1A1 (COL1A1), decorin, and dermatopontin, is significantly reduced in aged skin compared to its young skin. Yet, the majority of collagen mRNA expression levels of aged sun-exposed skin are similar to those found in young sun-exposed skin. This discrepancy is primarily attributable to a substantial decrease in collagen transcription in young sun-exposed skin, suggesting early molecular changes in matrisome transcription due to sun exposure, which preceded the emergence of clinical signs of photoaging. These findings shed light on the mRNA transcript profile of major matrisome proteins and their alterations in naturally aged and photoaged human skin, offering valuable insights into skin matrisome biology.

## 1. Introduction

Human skin is the most voluminous connective tissue in the human body. The bulk of human skin is composed of dense collagen-rich extracellular matrix (ECM), which is essential for the maintenance of skin structure, mechanical properties, and function [1]. Like all organs, the human skin undergoes a natural aging process as time passes. However, unlike other organs, the skin constantly experiences detrimental stress and damage from environmental factors like solar ultraviolet (UV) radiation. This leads to two distinct types of cutaneous aging: natural aging (intrinsic aging) and photoaging (extrinsic aging) [2]. Histological and ultrastructural examinations have pinpointed the primary changes in both naturally aged and photoaged skin within the dermal connective tissue. Biochemical evidence indicates that the prominent molecular feature of aging skin is characterized by fragmentation of collagen fibrils and declined collagen synthesis [3].

Collagenous ECM provides structural and mechanical stability to the skin and plays a crucial role in cell–ECM interactions. The collagenous ECM provides structural and mechanical support for appendages of the dermis including hair follicles, sebaceous glands, sweat glands, and vascular components. Age-related alteration to the dermal ECM creates a tissue microenvironment with many pathologic skin disorders, such as increased fragility [4], impaired vasculature support [5], poor wound healing [6,7], and promotion of cancer [8,9,10].

Recently, heterogeneous ECM components have been cataloged as matrisome [11,12]. The matrisome encompasses over 1000 proteins and is categorized into two main groups: the core matrisome and matrisome-associated proteins. The core matrisome is further divided into collagens, ECM glycoproteins, and proteoglycans, while the matrisome-associated proteins encompass ECM-affiliated proteins, ECM regulators, and secreted factors. In a recent study, we quantified matrisome proteins in naturally aged and photoaged human skin, revealing significant reductions of matrisomal proteins such as collagens (core matrisome), decorin (glycoprotein), and dermatopontin (proteoglycans) in both naturally aged and photoaged skin [13]. With aging, the activity of fibroblasts, the primary cells responsible for producing matrisomal ECM in the skin, decreases [1,13]. This is due to largely impaired transforming growth factor-beta (TGF-β) signaling, which is the major regulator of collagenous ECM synthesis [1,3]. The age-related decline in matrisomal proteins likely contributes to skin dermal aging. However, there are currently a limited number of published articles examining the human skin matrisome in the context of aging. One interesting question is whether these diminished matrisomal proteins at the protein level are attributed to reduced mRNA transcriptional levels in naturally aged and photoaged human skin. This study aims to address this question by evaluating the transcriptional levels of matrisomal proteins in both naturally aged and photoaged skin by comparing their expression in sun-protected versus sun-exposed skin.

## 2. Materials and Methods

**Core matrisome protein: collagen gene expression in young and aged human skin**. Skin biopsies were collected from healthy adult Caucasian volunteers with no history of skin inflammatory conditions or chronic diseases [13,14,15]. The volunteers were divided into two groups: a young group aged 22–30 years (mean age 26.7 ± 1.3 years, with 2 males and 4 females) and an aged group of 80 years or older (mean age 84.0 ± 1.7 years, with 2 males and 4 females). Additionally, skin biopsy samples were obtained from the hip region of healthy adults aged 48 ± 5 years (N = 6). The biopsies were taken from three anatomical sites: sun-protected areas (hip and underarm) and sun-exposed forearm (full-thickness samples, 4 mm in diameter). The skin biopsies were obtained during the winter month of January to eliminate any potential influence of recent ultraviolet radiation exposure on the skin samples. The biopsy sites were selected from non-pigmented areas of the skin. While the young skin samples exhibited a smooth texture without wrinkles, the aged skin samples displayed the presence of fine wrinkles. Photodamage of dorsal forearm skin was judged by wrinkles, pigmentation, skin turgor, and subject history of sun exposure [16]. All procedures involving human subjects were approved by the University of Michigan Institutional Review Board, and all subjects provided written informed consent before entering the study.

**RNA isolation and quantitative real-time RT-PCR**. Total RNA was extracted from skin, using a commercial kit (RNeasy midikit, Qiagen, Chatsworth, CA, USA), and reverse transcription was performed using Taqman Reverse Transcription kit (Applied Biosystems, Foster City, CA, USA). Real-time RT-PCR was performed using a Taqman Universal PCR Master Mix kit (Applied Biosystems, Foster City, CA, USA) and 7300 Real-Time PCR System (Applied Biosystems, Foster City, CA, USA). PCR procedures were performed using a robotic workstation (Biomek 2000; Beckman Coulter, Inc., Hialeah, FL, USA) to ensure accuracy and reproducibility. All primers and probes were purchased from Applied Biosystems (Assays-on-Demand™ Gene Expression Products). Multiplex PCR reactions contained primers and probes for the target gene and 36B4, a ribosomal protein used as an internal normalization control for quantitation.

**Immunohistology**. Immunohistology was performed as described previously [16]. Briefly, skin samples embedded in OCT were sectioned (7 µm), fixed in 2% paraformaldehyde, permeabilized with 0.5% Triton X-100 in phosphate-buffered saline (PBS), blocked with rabbit serum (5% in PBS), and incubated for one hour at room temperature with type I collagen and LAMA5 antibodies (Santa Cruz Biotechnology, Santa Cruz, CA, USA), followed by incubation with secondary antibody for one hour at room temperature. After staining, the slides were examined using a digital imaging microscope (Zeiss, Germany). Specificity of staining was determined by substituting isotype-control immunoglobulin (mouse IgG2a) for the primary antibodies. No detectable staining was observed with isotype-controls.Images were obtained using Zeiss microscopy and quantified by Image J (NIH). 

**Charts and statistics**. The data were organized in Microsoft Excel 365, and then transferred into GraphPad Prism (v.8) for statistical analysis and graph generation. All data are represented as Mean ± SEM. Statistical analysis was performed using GraphPad Prism (v.8) with unpaired two-sided Student’s *t*-tests, one-way analysis of variance (ANOVA) with Tukey’s method for multiple comparisons, or Kruskal–Wallis test with Dunn’s multiple comparisons test. Statistical significance was defined as *p* < 0.05. All experiments were repeated a minimum of three times unless otherwise stated. 

## 3. Results

### 3.1. Core Matrisome Protein: Collagen Gene Expression in Young and Aged Human Skin

Collagen represents by far the most abundant protein, accounting 90.1% of total matrisomal proteins in skin [13]. Among the identified collagens, we found 15 types. Type I collagen (COL1A1 and COL1A2) was the most abundant, comprising 89% of the total collagen content. In our present investigation, we aim to evaluate the mRNA transcriptional levels of these 15 collagens in both sun-protected and sun-exposed skin from both young and aged volunteers. To achieve this, skin punch biopsies were collected from a total of 12 volunteers, divided into two groups—young (<30 years, N = 6) and aged (>80 years, N = 6)—across three anatomical sites: sun-protected areas (hip and underarm) and sun-exposed forearm skin.

Initially, we assessed the gene expression of the 15 collagen types in adult human skin (average age: 48 ± 5 years, N = 6) and ranked them based on their abundance (Figure 1A). Analysis of gene expression by abundance revealed that COL1A1 displayed the highest expression, followed by COL4A1 and COL1A2, which is consistent with findings from our previously published proteomic analysis.

Subsequently, we compared collagen gene expression between young and aged human skin at each anatomical site. We noted a significant decrease in collagen gene expression in aged sun-protected hip (Figure 1B) and underarm (Figure 1C) skin compared to young hip and underarm skin, respectively. However, contrary to expectations, sun-exposed forearm skin of the aged did not exhibit a decrease in major collagens (COL1A1 and COL1A2) compared to young sun-exposed forearm skin (Figure 1D). Instead, several collagen types (COL12A1, 17A1, 14A1, 7A1) showed an increase in aged sun-exposed forearm skin.

Next, we conducted a comparison of collagen gene expressions across different anatomical sites. In young human skin, there is a notable reduction of the levels of major collagens (COL1A1, COL1A2, COL1A3) in the sun-exposed forearm compared to the sun-protected hip and underarm skin (Figure 1E). However, in aged sun-exposed forearm skin, the expression levels of major collagens (COL1A1 and COL1A2) remain unchanged, while some collagens (COL6A1, COL14A1, and COL17A1) exhibit an increase compared to sun-protected hip and underarm skin (Figure 1F).

Based on these data, we summarized the gene expression of major collagens: COL1A1 (Figure 1G), COL1A2 (Figure 1H), and COL1A3 (Figure 1I) in young and aged human skin at different anatomical sites (hip, underarm, and forearm). We observed that all major collagens are significantly reduced in aged skin from all three anatomical sites compared to young skin. In young human skin, major collagens are most highly expressed in the hip, followed by the underarm and forearm. Conversely, in young sun-exposed forearm skin, collagens are significantly reduced compared to sun-protected hip and underarm skin. However, in aged human skin, collagens do not exhibit significant changes among the hip, underarm, and forearm compared to young skin.

### 3.2. The Major Glycoprotein: Decorin Gene Expression in Young and Aged Human Skin 

Our previous proteomics investigation highlighted significant reduction of Decorin (DCN), a predominant glycoprotein in human skin, in aged skin compared to young human skin [13]. We assessed DCN mRNA expression in both young and aged human skin. Consistent with our proteomics findings, DCN mRNA expression was found to be decreased in aged skin compared to young skin in both sun-protected and sun-exposed areas (Figure 2). Additionally, when comparing DCN mRNA expression across different anatomical sites of the skin, it was observed that DCN gene expression is reduced in the sun-exposed forearm compared to the sun-protected area in both young and aged skin.

### 3.3. The Major Proteoglycans: Dermatopontin Gene Expression in Young and Aged Human Skin

We reported that the dermatopontin (DPT) protein, a primary proteoglycan in human skin, undergoes a significant reduction of aged compared to young skin [13]. Consistent with this, we observed a decrease in DPT gene expression in aged skin compared to young skin, in both sun-protected and sun-exposed skin (Figure 3). Furthermore, when comparing DPT gene expression across different anatomical sites, we noted a reduction of expression levels in the sun-exposed forearm compared to sun-protected areas in both young (Figure 3) and aged skin (Figure 3).

### 3.4. Basement Membrane Protein Gene Expression in Young and Aged Human Skin 

Our previous proteomic analysis revealed that many basement membrane components were significantly reduced with age [13]. We identified a total of 10 basement membrane proteins from proteomic analysis. We examined these 10 basement membrane proteins’ mRNA expression in young and aged human skin. Only a decrease in LAMA5 was seen in aged skin compared to young skin in both sun-protected hip (Figure 4A) and underarm (Figure 4B), and sun exposed forearm skin (Figure 4C). Similarly, comparing basement membrane protein gene expression from different anatomic site indicated that only LAMA5 is decreased in sun-exposed forearm skin, compared to sun-protected skin (hip and underarm) in both young (Figure 4D) and aged (Figure 4E) human skin.

### 3.5. Reduced Expression of Type I Collagen and LAMA5 in Aged Human Skin: Immunohistology

Subsequently, we verified the age-related decrease in expression levels of type I collagen and LAMA5 by immunostaining. Consistent with the mRNA expression data, type I collagen protein expression was reduced in aged compared to young skin in both sun-protected skin (hip and underarm) (Figure 5A lower left and middle bars) and sun-exposed forearm (Figure 5A lower right panel) skin. In young human skin, type I collagen protein expression is notably lower in sun-exposed forearm compared to the sun-protected hip and underarm (Figure 5A upper right bar) skin. This pattern is not as evident in aged human skin (Figure 5A right lower bar). Similarly, the protein expression of LAMA5 was significantly decreased in aged compared to young skin in both sun-protected hip (Figure 5B lower left and middle bars) and sun-exposed forearm (Figure 5B lower right bar) skin. LAMA5 protein expression is reduced in sun-exposed forearm skin compared to sun-protected skin (hip and underarm) in both young (Figure 5B upper right bar) and aged (Figure 5B lower right bar) skin.

## 4. Discussion

Human skin experiences notable changes due to the natural aging process coupled with UV radiation from the sun (photoaging). The decline in matrisomal proteins contributes to the thinning and fragility of aged skin. Our study demonstrates a significant decrease in the overall levels of core matrisomal proteins, in both mRNA and protein expression, in aged sun-protected skin in comparison to young sun-protected skin. Notably, the levels of type I collagen, the predominant matrisomal protein in the skin, significantly reduced in both mRNA and protein expression in aged, sun-protected skin, when compared to its young sun-protected skin. These findings imply that decreased transcriptional activity leads to age-related decline in major core matrisomal proteins.

In contrast to sun-protected skin, there is no noticeable alteration in the transcription of type I collagen in aged sun-exposed skin, as compared to young sun-exposed skin. This discrepancy is primarily attributable to a substantial decrease in type I collagen transcription in sun-exposed young skin, as illustrated in Figure 1I. Furthermore, the gene expression of the primary glycoprotein decorin and the proteoglycan dermatopontin are also diminished in young sun-exposed skin compared to sun-protected skin (Figure 2 and Figure 3).

These results are consistent with our previous proteomic study, where we observed a notable decrease in core matrisomal proteins in young sun-exposed forearm skin [13]. This discovery is particularly intriguing because it indicates molecular alterations in matrisomal proteins occurring in young, sun-exposed forearm skin, even in the absence of visible signs of photoaging. These results suggest that exposure to sunlight in young adult skin may induce noteworthy molecular changes in matrisomal proteins. While youth is typically associated with a minimal risk of UV-induced skin damage, our data suggest that molecular alterations related to photoaging in matrisomal proteins are already apparent, potentially signaling an increased risk of long-term detrimental effects later in adulthood. Several epidemiological studies have highlighted the significant contribution of UV exposure during childhood and adolescence to cumulative UV exposure over a lifetime [17,18,19]. It is estimated that approximately 40–50% of total UV exposure by age 60 occurs before the age of 20 [17]. This vulnerable period in young adult could serve as a window for the onset of long-term adverse effects of UV exposure, as evidenced by the changes observed in matrisomal proteins. Hence, the implementation of effective UV protection from a young age may be imperative to mitigate the risk of long-term damage to the skin caused by photoaging.

Human skin fibroblasts exhibit considerable heterogeneity, primarily because they do not all originate from the same embryonic lineage [20,21]. Skin fibroblasts from different anatomical sites displayed distinct and characteristic transcriptional patterns [22]. This heterogeneity is particularly crucial during wound healing and scar formation [23]. During wound healing, fibroblasts from different anatomical sites exhibit varying responses to growth factors and cytokines, influencing the outcomes of wound healing, scarring, and tissue regeneration. In our study, we observed no significant difference in the major matrisome transcription between the hip and underarm, which are two sun-protected anatomical sites, in both young and aged skin. We also observed no changes in type I collagen transcription among different anatomic sites regardless of sun-exposed or sun-protected skin in aged human skin. Instead, several collagen transcriptions are increased in aged sun-exposed forearm skin, suggesting that the collagen gene expression is regulated differently in aged sun-exposed skin compared to sun-protected skin. One explanation is that sun-exposed skin has damaged dermis such as fragmentations of collagen and elastosis. The alterations in dermal matrisome due to sun exposure could have a considerable influence on the interactions between dermal fibroblasts and the surrounding collagen fibrils, thereby significantly affecting the regulation of collagen by fibroblasts. Regardless of sun exposure, the mRNA expression level of the main proteoglycan DCN demonstrates a significant decrease in aged skin compared to young skin. These findings align with our prior observation that DCN is a predominant proteoglycan in skin and is diminished in aged human skin [13,24]. DCN is a small leucine-rich proteoglycan that binds to specific sites on the surface of type I collagen fibrils. This interaction is crucial for the proper assembly of collagen fibrils. Deficiency in DCN is believed to contribute to skin abnormalities such as bruising, hyperelasticity, and Ehlers-Danlos syndrome, a group of hereditary connective tissue disorders [25,26]. Therefore, the age-related reduction of DCN may potentially affect the quality of dermal collagen fibrils in aged skin.

Additionally, DPT, the most abundant glycoprotein, is also significantly reduced in aged skin, compared to young skin in both sun-protected and sun-exposed skin. DPT is one of the non-collagenous components of the ECM and is found in various tissues [27]. Despite its abundance in the dermis, the knowledge about DPT and its role in skin remains limited. It has been reported that DPT interacts with decorin to modulate collagen fibrillogenesis [28,29,30]. DPT-knockout mice demonstrate an Ehlers-Danlos syndrome-like phenotype, such as skin fragility and impaired lateral arrangement of collagen microfibrils [31,32]. DPT is also involved in cell–matrix interactions and matrix assembly. Proteoglycans act in concert with glycoproteins to regulate ECM assembly, ECM-cell interaction, and signaling. Although dermal proteoglycans and glycoproteins are present in much lower abundance than collagen, evidence indicates that these molecules are important in the physiology of skin [31]. For example, the small proteoglycan decorin binds to type I collagen, and targeted disruption of decorin results in aberrant collagen fibrils and in a reduction of the tensile strength of skin [27]. Clearly, investigating the expression and functions of DCN and DPT in skin ECM biology and aging is of considerable interest.

Basement membranes constitute specialized ECM thin layers situated at the junctions of the epidermis and dermis, as well as beneath the endothelial cells in blood vessel walls [33]. Our focus was on LAMA5, a component of these membranes, which we observed to exhibit decreased expression in aged skin compared to young skin, regardless of sun exposure. When comparing gene expression of basement membrane proteins across various anatomical sites, we found that LAMA5 alone showed decreased expression in sun-exposed forearm skin compared to sun-protected skin, in both young and aged skin. Lamins (LM) are composed of a, β, and c chains that assemble into aβc heterotrimers to form 16 different isoforms [34]. The LAMA5 chain is one of the major α chains and plays a crucial role in the maintenance of epithelial basement membranes’ integrity [35,36]. LAMA5 knockout in mice results in embryonic lethality and manifests severe developmental abnormalities in multiple tissues, such as extracerebral malformations, as well as aberrant lung, kidney, tooth, and hair follicle development [34,37]. LAMA5 is intriguing due to its consistent reduction of both natural and photoaging, observed at both mRNA and protein levels. However, the mechanisms governing LAMA5 gene regulation remain poorly understood. This regulation likely entails a complex interaction of molecular processes, such as transcriptional and post-transcriptional regulation, as well as epigenetic modifications. Clearly, delving into the regulation of the LAMA5 gene in the context of skin aging holds significant interest.

The integrity of dermal structure and mechanics is vital for skin function. Skin structural and mechanical properties are determined essentially by matrisomal proteins. Our findings, combined with previous research, clearly demonstrate a decline in matrisome components as we age. Specifically, key matrisome proteins such as collagen, proteoglycans, and glycoproteins are significantly reduced in aged skin, both at the mRNA and protein levels. These results shed light on the changes that occur in matrisome proteins during natural aging and photoaging of human skin, providing valuable insights into the field of skin matrisome biology.

## Figures and Tables

**Figure 1 biomolecules-14-00900-f001:**
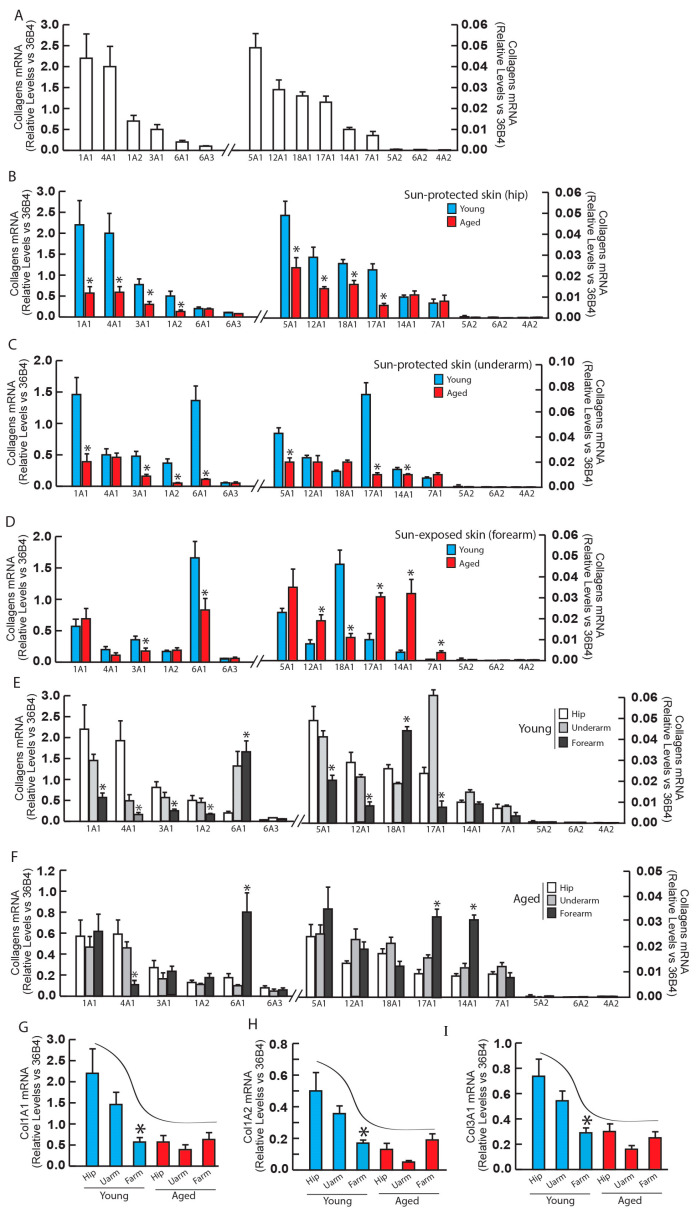
Core matrisome protein: collagen gene expression in young and aged human skin. Skin punch biopsies were collected from young (<30 years, N = 6) and aged (>80 years, N = 6) sun-protected areas (hip and underarm) and sun-exposed (forearm) skin. Total RNA was extracted, and collagen mRNA levels were quantified by real-time RT-PCR and were normalized by the housekeeping gene (36B4, internal control). (**A**) Collagen mRNA expression by their abundance in adult human skin (48 ± 5 years, N = 6). (**B**) Collagen mRNA expression in young and aged sun-protected hip skin. (**C**) Collagen mRNA expression in young and aged sun-protected underarm skin. (**D**) Collagen mRNA expression in young and aged sun-exposed forearm skin. (**E**) Collagen mRNA expression in young skin at different anatomical sites (hip, underarm, and forearm). * *p* < 0.05 sun-exposed forearm vs. sun-protected hip and underarm. (**F**) Collagen mRNA expression in aged skin at different anatomical sites (hip, underarm, and forearm). * *p* < 0.05 sun-exposed forearm vs. sun-protected hip and underarm. (**G**) COL1A1 mRNA expression in young vs. aged skin at different anatomical sites (hip, underarm, and forearm). (**H**) COL1A2 mRNA expression in young vs. aged skin at different anatomical sites (hip, underarm, and forearm). (**I**) COL1A3 mRNA expression in young vs. aged skin at different anatomical sites (hip, underarm, and forearm). Mean ± SEM. * *p* < 0.05.

**Figure 2 biomolecules-14-00900-f002:**
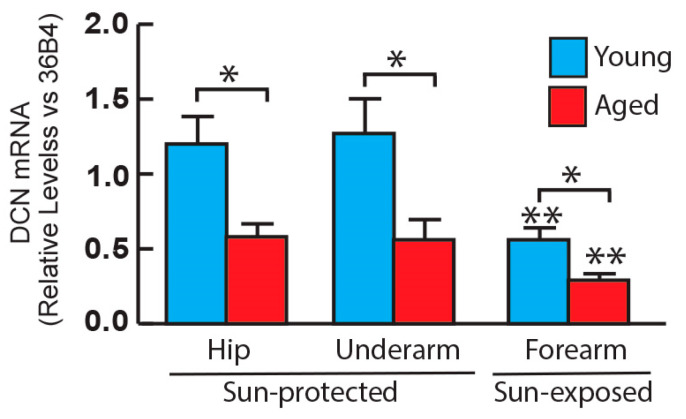
The major glycoprotein: Decorin gene expression in young and aged human skin. Skin punch biopsies were collected from young (<30 years, N = 6) and aged (>80 years, N = 6) sun-protected areas (hip and underarm) and sun-exposed (forearm) skin. Total RNA was extracted, and decorin mRNA levels were quantified by real-time RT-PCR and were normalized by the housekeeping gene (36B4, internal control). Mean ± SEM. * *p* < 0.05 young vs. aged skin. ** *p* < 0.05 sun-exposed forearm vs. sun-protected skin (hip and underarm) in both young and aged individuals, respectively.

**Figure 3 biomolecules-14-00900-f003:**
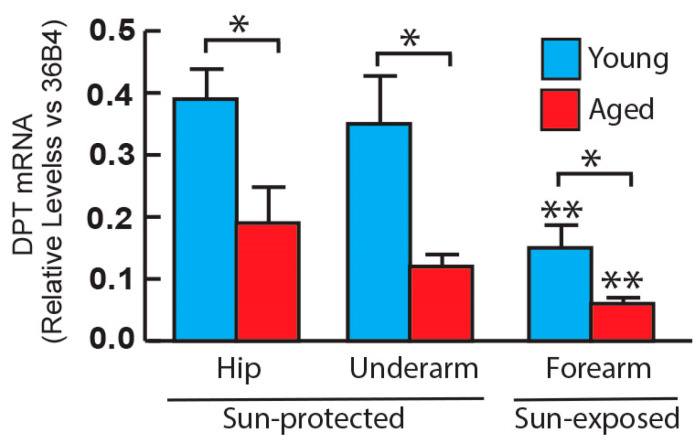
The major proteoglycans: dermatopontin gene expression in young and aged human skin. Skin punch biopsies were collected from young (<30 years, N = 6) and aged (>80 years, N = 6) sun-protected areas (hip and underarm) and sun-exposed (forearm) skin. Total RNA was extracted, and dermatopontin mRNA levels were quantified by real-time RT-PCR and were normalized by the housekeeping gene (36B4, internal control). Mean ± SEM. * *p* < 0.05 young vs. aged skin. ** *p* < 0.05 sun-exposed forearm vs. sun-protected skin (hip and underarm) in both young and aged individuals, respectively.

**Figure 4 biomolecules-14-00900-f004:**
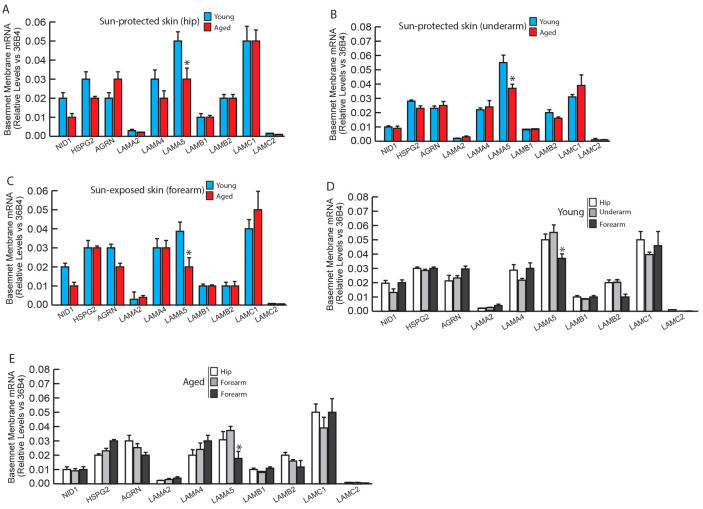
Basement membrane protein gene expression in young and aged human skin. Skin punch biopsies were collected from young (<30 years, N = 6) and aged (>80 years, N = 6) sun-protected areas (hip and underarm) and sun-exposed (forearm) skin. Total RNA was extracted, and basement membrane mRNA levels were quantified by real-time RT-PCR and were normalized by the housekeeping gene (36B4, internal control). (**A**) Basement membrane mRNA expression in young and aged sun-protected hip skin. (**B**) Basement membrane mRNA expression in young and aged sun-protected underarm skin. (**C**) Basement membrane mRNA expression in young and aged sun-exposed forearm skin. (**D**) Basement membrane mRNA expression in young sun-protected (hip and underarm) vs. sun-exposed forearm skin. (**E**) Basement membrane mRNA expression in aged sun-protected (hip and underarm) vs. sun-exposed forearm skin. Mean ± SEM. * *p* < 0.05.

**Figure 5 biomolecules-14-00900-f005:**
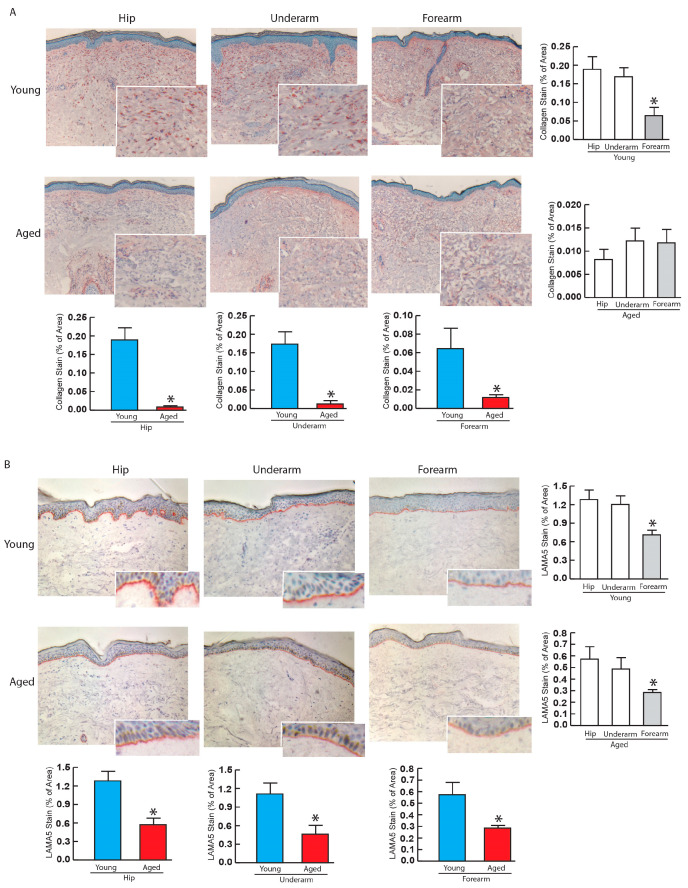
Type I collagen and LAMA5 immunohistology in young and aged human skin. Skin punch biopsies were collected from young (<30 years, N = 6) and aged (>80 years, N = 6) sun-protected (hip and underarm) and sun-exposed (forearm) skin, and embedded in OCT. (**A**) Type I collagen protein expression in young sun-protected hip (left panel) and underarm (middle panel) and sun-exposed forearm (right panel) skin. (**B**) LAMA5 protein expression in young sun-protected hip (left panel) and underarm (middle panel) and sun-exposed forearm (right panel) skin. Specificity of staining type I collagen and LAMA5 staining were quantified by Image J (NIH) and expressed by % of specific staining area. Mean ± SEM. * *p* < 0.05.

## Data Availability

Data is contained within the article.

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
