# Peer review of "Alterations of Matrisome Gene Expression in Naturally Aged and Photoaged Human Skin In Vivo"

_biomolecules, 2024, doi:10.3390/biom14080900_

Round 1

Reviewer 1 Report

Comments and Suggestions for Authors

The study investigated the mRNA expression of core matrisome proteins in human skin, comparing young vs. aged and sun-protected vs. sun-exposed skin.

It was found that the core matrisome transcription was decreased in aged skin.

In addition, it was observed that the collagen mRNA levels in aged sun-exposed skin are similar to young sun-exposed skin. Also, it seems that in sun-exposed areal the mRNA levels of the major glycoproteins as decorin, dermatopontin, were reduces in young and aged skin.

The authors suggested that early molecular changes due to sun exposure before visible signs of photoaging appear.

However, due to the following problems and lack of information, the conclusions of this study are very limited.

1. There is a lack of information about the volunteers:

Lack of data on wrinkles, pigmentation, skin turgor and sun exposure of volunteers, both old and young skin.

Please, show the data.

 Possible correlations, e.g. age vs. mRNA collagen or pigmentation vs. mRNA (in the young group), should be analyzed here, which could provide further information.

2. there is no indication of when and how the last sun exposure was (summer, winter, T-shirt, use of sunscreen, etc.), which is very important for the interpretation of the results for the forearm. Therefore, it is quite possible that you measured acute effects of sunlight on dermal mRNA, which could also very well explain the reported results.

Please provides the missing information and discuss

3. without control groups for the old and young forearm groups, i.e. subjects with forearms without sun exposure, it is quite possible that the differences between the mRNA levels depend more on the anatomical site than on the sun exposure. Here, the results from hip and forearm, both sun-protected, already showed many differences, e.g. Fig. 1E: collagen 1A1 hip about 2.2x and 1F forearm only about 1.5x.

 Please, discuss this issue and provide further evidence that the anatomical location does not have such a large effect on the mRNA level

Please, rearrange the figures, e.g. the results of Fig. 1E and Fig. 1F could be better presented in one figure. Similarly, Fig. 1G and Fig. 1H

4 Fig. 2 and 3: In both figures, B and C do not provide more information than already shown in A.

5 Figure 4 only shows the results for the hip and forearm. What happened to the results for the forearm?

6 Figure 5 also only shows the results for the hip and forearm. What happened to the results for the underarm? In my opinion, the comparison between the underarm and forearm is more meaningful because the skin of the hip is more different from the skin of the forearm than the skin of the underarm.

In summary, it can be said that although the data presented is interesting, it cannot be interpreted and discussed in the way it was done in the paper without additional information and experiments. 

Reviewer 2 Report

Comments and Suggestions for Authors

Manuscript titled "Alterations of matrisome gene expression in naturally aged and 1 photoaged human skin in vivo" investigate  the mRNA expression of the some matrisome proteins in human skin, comparing young versus aged and sun-protected versus sun-  exposed skin.  The subject of the manuscript is very interesting and meets the expectations related to the need to inhibit the process of skin collagen degradation. 

My comments on the work:

In the Abstract-please briefly describe the methods used

Introduction- this part of the work lacks an explanation of why there is/may be a decrease in matrisome mRNA and proteins.

Have such tests been performed before? on other tissues? On humans, on animals?

age 30 is stil young?

Results: How were the 15 types of collagen identified?

How many  volunteers were there? in Materials and Methods it talks about 6 at the age of about 30 and 6 at the age of about 80, the results show an additional 6 at the age of 48????

line 290- the conclusion is too far-reaching. The subjects analyzed were volunteers around 30 years of age, no children

line 301- there are no citations given confirming this statement

Could the authors present in a short summary description how much mRNA and proteins change in people around 30 and 80 years of age?

What do the authors consider the main novelty of their work?
